# Development of Paper Actuators Based on Carbon-Nanotube-Composite Paper

**DOI:** 10.3390/molecules26051463

**Published:** 2021-03-08

**Authors:** Takahiro Ampo, Takahide Oya

**Affiliations:** Graduate School of Engineering Science, Yokohama National University, Yokohama 240-8501, Japan; oya-takahide-vx@ynu.ac.jp

**Keywords:** carbon nanotubes, actuator, paper, carbon-nanotube-composite paper, ionic liquid, carbon powder

## Abstract

We propose a unique soft actuator—a paper actuator—based on carbon-nanotube-composite paper (CNT-composite paper), which is a composite of carbon nanotubes (CNTs) and paper. CNT-composite paper has highly efficient properties because of the contained CNTs, such as high electrical conductivity and semiconducting properties. We are considering using CNT-composite paper for various devices. In this study, we successfully developed a paper actuator. We determined the structure of the paper actuator by referencing that of bucky-gel actuators. The actuator operates using the force generated by the movement of ions. In addition to making the paper actuator, we also attempted to improve its performance, using pressure as an index and an electronic scale to measure the pressure. We investigated the optimal dispersant for use in paper actuators, expecting the residual dispersant on the CNT-composite paper to affect the performance differently depending on the type of dispersant. Referring to research on bucky-gel actuators, we also found that the addition of carbon powder to the electrode layers is effective in improving the pressure for paper actuators. We believe that the paper actuator could be used in various situations due to its ease of processing.

## 1. Introduction

Carbon nanotubes (CNTs) are substances composed of carbon. CNTs have properties that are beneficial for technology such as high electrical conductivity, chemical stability, light weight, flexibility, and high strength [1,2,3,4]. Due to these characteristics, CNTs are expected to be utilized in various fields. However, CNTs are nanoscale substances and exist in powder form. Therefore research is being conducted not only on simply using CNTs alone but also on applying them to devices in the form of composite materials that combine CNTs with other materials. We focus on paper, which is an easy-to-process and familiar material, as the other component of a composite material. Combining CNTs and paper, the composite material has both of these materials’ characteristics. We call this composite material “CNT-composite paper” [5], and we are studying its application to various devices.

Devices called actuators convert input energy into physical motion. Many mechanical actuators can move with electric energy. Currently, metal actuators are mainstream. Recently, wearable devices have been attracting attention in areas such as the medical field. Wearable soft actuators require features that differ from those of conventional actuators, such as light weight and flexibility. CNTs are attracting attention as one potential material for these soft actuators. CNT-based soft actuators using heat [6], light [7], or ion bias [8,9,10] have been reported, but are not yet commonly used. One of the reasons for this is that these devices’ movement is simpler than that of existing actuators. We hope to solve this problem with a paper actuator made of CNT-composite paper. Because it would be made of paper, complicated three-dimensional structures could be formed relatively easily, and movement with a high degree of freedom would be possible. For these reasons, we aim to develop a paper actuator using CNT-composite paper.

## 2. Results

In this study, we successfully developed paper actuators using CNT-composite paper. An evaluation of our first paper actuator is described in Section 2.1. The performance of the first sample was insufficient, so we attempted to improve it. Section 2.2 describes the results of these efforts.

### 2.1. First Actuator using CNT-Composite Paper

We believed the characteristics of CNTs, such as their high electrical conductivity and light weight, and the characteristics of paper, such as its light weight and flexibility, to be suitable for soft actuators. Therefore, we considered the development of paper actuators using CNT-composite paper, which has the characteristics of both CNTs and paper. The actuator’s structure was determined with reference to bucky-gel actuators [8,9,10], which are soft actuators that have already been developed. Bucky-gel actuators utilize the force generated by the movement of ions. Figure 1 shows the structure of the CNT-composite paper actuator we made, and Figure 2 shows the schematic operation of the actuator.

Ionic liquids are used, so the ions are spread over the entire device. When a voltage is applied to this device, ions move to each electrode layer. The sizes of the cations and anions are different [11], so the electrode layer to which the large ions have moved expands, and the electrode layer to which the small ions have moved shrinks. As a result, the actuator is deformed.

We actually made a paper actuator and measured the pressure it generated. We made CNT-composite paper with 30 mg of CNT and 100 mg of pulp for a circle with a diameter of 6 cm. Figure 3 shows an example of a prepared CNT-composite paper sample. We then cut the circular CNT-composite paper into two rectangular sheets measuring 1.5 × 4 cm^2^ for use in the actuator. After that, we used these two sheets of CNT-composite paper to make the actuator in accordance with the procedure given in Section 4.3. Figure 4 shows the actuator.

For the first test, we used an electronic scale to measure the pressure the device generated. The pressure with which the actuator pushes the scale was used as an index. We will explain the specific measurement method in Section 4.4. Figure 5 shows the measured generated pressure as a function of time. Here, we represent the results of a sample using 100 mg of catechin as a dispersant. The resistance of this sample was 12.2 Ω/sq.

Figure 5 shows that when a voltage is applied, the pressure generated by the actuator increases and converges to an almost constant value. From this, we determined that the actuators using CNT-composite paper can move like bucky-gel actuators.

### 2.2. Examination to Improve Performance of Paper Actuator

We confirmed the operation of the CNT-composite paper actuator in Section 2.1. However, the generated pressure is weak and must be increased for practical reasons. This subsection describes several approaches to increasing the generated pressure.

#### 2.2.1. Examination of the Type of Dispersant

CNTs must be isolated and dispersed in order to be fully utilized. Ultrasonic irradiation is sometimes used to isolate and disperse CNTs in water. In these instances, a dispersant is used to prevent the reaggregation of CNTs. This dispersant is difficult to separate from the CNTs even when CNTs are used as part of a composite material, and the presence of the dispersant significantly affects the characteristics of the composite material. We therefore investigated three compounds with regard to their suitability for use as the dispersant in a CNT-composite paper actuator. Specifically, we compared the performance obtained using sodium dodecyl sulfate (SDS) as an anionic surfactant, polysorbate 20 as a nonionic surfactant, and catechin as a non-surfactant. The structure of each dispersant used in this study is shown in Figure 6.

Figure 7 shows the generated pressure as a function of time when conditions other than the dispersant were the same.

We consider the relationship between the results in Figure 7 and the dispersion force of the dispersant. We thought that the resistance value of CNT-composite paper would be lowered if a dispersant with strong dispersion force was used. This is because if the dispersion force is strong, the CNTs will be dispersed uniformly in the dispersion. Therefore, if we could use such dispersion to make the CNT-composite paper, the CNTs could construct uniform networks in the composite paper. We anticipated that, as a result, this would create a composite paper with a lower resistance value. From the relationship between dispersion force and resistance value, we evaluated the dispersion force of each dispersant by measuring the resistance value of the CNT-composite paper. Table 1 shows the resistance values of the CNT-composite paper using each of the three types of dispersant.

From the results in Table 1, we can infer that the dispersion force goes from stronger to weaker in the order of SDS to catechin to polysorbate 20. From this and the results in Figure 7, we can see that the pressure generated by the sample using SDS with strong dispersion force is smaller than that generated by the other samples. This result is considered to be caused by SDS, which is an ionic surfactant, inhibiting the movement of ions. Another considerable factor is that the space in the paper increased due to the bundled CNTs that remained undispersed because of the weak dispersion force. We believe that bundled CNTs assist the movement of ions by creating space in the network of CNTs as spacers, i.e., a certain amount of bundled CNTs is required in the composite paper to create the paper actuator. We believe that this is way the generated pressure increased in the samples using catechin and polysorbate 20.

In addition, we measured the difference in the degree of dispersion depending on the type of dispersant using a scanning electron microscope. It was difficult to determine the dispersed state of CNTs in the CNT-composite paper. Therefore, we measured the dispersions in a state in which they were dropped on silicon substrates and dried. The specific measurement method is shown in Section 4.5. The measurement results are shown in Figure 8.

The results in Figure 8 show that the CNT bundles tended to be thicker when polysorbate 20 was used as opposed to SDS. We also found that there was little difference in the thickness of the CNT bundles when catechin was used compared with when SDS was used. However, we found some CNT lumps on the catechin side. We expect that these results will be the same for CNT-composite paper as well. We believe these factors to be due to the difference in the dispersion force of the dispersants.

#### 2.2.2. Examination of Addition of Carbon Powder

From the results in Section 2.2.1, we confirmed that the dispersion state of CNTs differs depending on the type of dispersant. We expect that this dispersed state will be the same in CNT-composite paper. In addition, we found that the pressure generated by poorly dispersed samples tends to be high. From these findings, we expected that thick bundles and lumps of CNTs would increase the pressure. We presumed that thick bundles and lumps of CNTs would create space inside the CNT-composite paper and help the movement of ionic liquids. As a principle similar to this, it has been reported that the performance of bucky-gel actuators can be improved by adding carbon powder to increase the space in the gel [12]. Considering these findings, we investigated whether the performance of the CNT-composite paper actuator could be improved by adding carbon powder to the CNT-composite paper. We did this by comparing three types of samples: one with 30 mg of CNT (Sample 1), one with 60 mg of CNT (Sample 2), and one with 30 mg of CNT and 30 mg of carbon powder (Sample 3), each on a circular piece of CNT-composite paper 6 cm in diameter. We standardized the pulp volume for all samples to 100 mg and used the preparation procedure described in Section 4.1 and Section 4.2. The measured results are shown in Figure 9.

First, we discuss the relationship between resistance and pressure for the results shown in Figure 9. Table 2 shows the resistance values of each CNT-composite paper used in each sample.

From Table 2, we can see that the resistance value goes from lower to higher in the order of Sample 2 to Sample 3 to Sample 1. The results in Figure 9 and Table 2 show that Samples 2 and 3 exhibited higher pressure values than Sample 1 with a large resistance value. This is because the range in which ion movement occurs becomes wide in Samples 2 and 3. When the resistance of CNT-composite paper is low, the area where a voltage large enough to move ions is applied widens. Therefore, the number of ions that affect the deformation increases, as does the pressure.

## 3. Discussion

We aimed to develop a paper actuator using CNT-composite paper. First, we determined the structure referencing bucky-gel actuators and achieved the operation of the CNT-composite paper actuator in the same manner. The performance of the actuator was evaluated using an electronic scale to measure pressure. We confirmed that when a voltage was applied, the pressure increased and converged to a constant value. Therefore, we found that the CNT-composite paper actuator can also operate using the movement of ions. In addition, the displacement did not decrease during the 10 min the voltage was applied, so we consider this device to be capable of withstanding long-term use. Next, we tried to increase the generated pressure as one way to improve the performance of the actuator.

We investigated the effect of the dispersant used when preparing the CNT dispersion. Generally, the dispersant used in the CNT dispersion liquid remains even when the CNT is made into a composite material; it is difficult to remove the dispersant. The residual dispersant also affects the performance of the composite material. This is also true for CNT-composite paper. Therefore, we considered the types of dispersant suitable for use in CNT-composite paper actuators. In this study, we prepared three CNT-composite paper actuators using different types of dispersant and compared their generated pressures. The dispersants used were SDS, polysorbate 20, and catechin. We found that the resistance of CNT-composite paper went from lower to higher in the order of SDS to catechin to polysorbate 20. However, we determined that the pressure went from higher to lower in the order of catechin to polysorbate 20 to SDS. From this, we speculate that if an electrode layer has low resistance, the pressure will increase, and that the ionic surfactant will inhibit the movement of ions and decrease the pressure. In addition, we believe that the space that the undispersed bundled CNTs create inside the paper helps the movement of ions.

Next, we investigated the effect of adding carbon powder to the electrode layers. This was done to further examine the effect of bundled CNTs creating space inside the paper when examining dispersants. The structure of the CNT-composite paper actuator developed in this study was based on that of bucky-gel actuators. A study on bucky-gel actuators found that their displacement could be improved by adding carbon powder to the electrode layers. For our study, the factors that improve displacement are considered as follows. The addition of carbon powder increases the space inside the electrode layers. The increased space assists the movement of the ionic liquid by increasing the number of ions that affect the deformation, thereby increasing the displacement of the actuator. CNTs exist at a nanometer scale, while carbon powder exists at a micrometer scale. We speculated that the larger the grain, the greater the effect of spreading inside of the CNT-composite paper. Therefore, we thought that adding carbon powder to the electrode layers would further improve the displacement. We confirmed that the addition of carbon powder increases the pressure. This showed that the addition of carbon powder to the electrode layers is effective for improving the pressure in the CNT-composite paper actuators. In addition, we measured the pressure when the same amount of CNT as carbon powder was added to the electrode layers. As a result, we found that the pressure increased significantly when carbon powder was applied to the electrode layers. From this, we speculate that carbon powder with larger grains creates more space inside the CNT-composite paper.

Finally, we found that the pressure generated when we measure similar resistance samples that we made under the same conditions was different. We attribute this to the randomness of the internal structure of the CNT-composite paper. We made CNT-composite paper by hand for this study. Thus, we anticipate that the internal structure of the CNT-composite paper will vary even when produced under similar conditions. We believe that different pressures were obtained due to the difference in the internal structure of CNT-composite paper. We view such variations in the production process as a subject for future study.

## 4. Materials and Methods

This section describes how to make the CNT-composite paper actuator and how we evaluated samples.

### 4.1. How to Make CNT-Composite Paper

To produce CNT-composite paper, we used an ancient papermaking method for producing Japanese *washi* paper. The manufacturing procedure is given below:We stir the pulp into pure water to prepare the pulp dispersion.We prepare a CNT dispersion liquid by mixing and dispersing CNT and catechin, a dispersant, in pure water using ultrasonic treatment.We mix the pulp dispersion and CNT dispersion to prepare the pulp/CNT mixture.We place a circular mold on a fine mesh and pour the pulp/CNT mixture into the mold.We dry and mold the deposits on the mesh with hot pressing.

In this study, 30 mg of single-walled CNT (ZEONANO SG101, ZEON Corporation, Tokyo, Japan) with an average diameter of 3 to 5 nm and a length of 100 to 600 μm and 100 mg of pulp were used for one sheet of the CNT-composite paper. The prepared sample was a circle with a diameter of 6 cm. The procedure is shown schematically in Figure 10.

### 4.2. Resistance Value Measurement of CNT-Composite Paper

We used the sheet resistance value as a performance evaluation index for the CNT-composite paper. We used the commonly used four-probe method to measure the sheet resistance value and we used a semiconductor parameter analyzer (KEITHLEY, 4200A-SCS, Solon, OH, USA) as the measuring instrument.

### 4.3. How to Make the CNT-Composite Paper Actuator

The procedure for making the actuator is shown below:
(1)We use the papermaking method described in Section 4.1 to make ordinary paper that does not contain CNTs. We used 30 mg of pulp to make one 1.5 × 4 cm^2^ sheet.(2)We moisten both of the two sheets of CNT-composite papers made with the method in Section 4.1 used for the electrode layers and the pulp paper used for the electrolyte layer.(3)We sandwich the paper used for the electrolyte layer between the two sheets of CNT-composite paper.(4)We bond the sheets of paper together with heat pressing.(5)We drop an ionic liquid on the bonded stack and let it penetrate the whole paper stack. We used 100 μL of ionic liquid (EMI-TFSI, TOYO GOSEI CO., LTD., Tokyo, Japan) for each sample.

Figure 11 shows a schematic of the procedure described above. Table 3 shows the structural formula and volume of the cations and anions in the ionic liquid used.

### 4.4. Measurement of the Generated Pressure of the Actuator

We used the magnitude of the pressure generated when a voltage was applied as an index of the performance of the manufactured actuator. For simplicity, we used an electronic scale for the measurement. We expected the actuator to change the value of the electronic scale by pushing a piece of plastic placed on the electronic scale. We considered that this value could be regarded as the pressure generated by the actuator. Therefore, we derived the pressure from the value measured by the electronic scale. First, we placed a 1.5 × 1.5 cm^2^ piece of plastic on the electronic scale. Next, we placed the actuator on a piece of plastic with the cathode side facing down. Then, we initialized the output of the scale to show zero. We then assumed that the value displayed when the voltage was applied was the pressure generated by the actuator and converted it to the pressure value using Equation (1):Pressure [Pa] = Scale value [kg]/Standard gravity [m/s^2^]/Area [m^2^](1)

We graphed the data consequently obtained in the form of the generated pressure with respect to time and evaluated it as the performance of the actuator. We set the applied voltage to 2 V and the application time to 10 min for the measurement.

### 4.5. Examination of the Effect of the Type of Dispersant

We compared three dispersants to investigate the effect of dispersant type on paper actuators. The dispersants were SDS, polysorbate 20, and catechin. We used 75 mg of SDS, 10 μL of polysorbate 20, or 100 mg of catechin for 30 mL of pure water when preparing the CNT dispersion used in the procedure described in Section 4.1. We set the amount of each dispersant so that even if the amount was increased further, almost no decrease in resistance was confirmed. We made the CNT-composite paper in accordance with the procedure given in Section 4.1 without changing anything other than the amount of the dispersants. Then, after measuring the resistance value of each CNT-composite paper produced in accordance with the procedure in Section 4.2, we made the actuator using the procedure described in Section 4.3. We measured the generated pressure of each actuator using the procedure given in Section 4.4 and compared the performance. In this manner, we examined the effect of each dispersant on the CNT-composite paper actuator. In addition, we observed and evaluated the dispersion state of the dispersion with a scanning electron microscope (VE-8800, KEYENCE, Osaka, Japan). This experiment was to clarify the relationship between the dispersion force of the dispersant and the generated pressure. Measuring the dispersion state in the CNT-composite paper would be preferable but doing so was difficult for us because of the pulp and its network in the paper. Therefore, we attempted measuring the dispersion state with the dispersions dropped on silicon substrates. We prepared the CNT dispersions under the same conditions as when we used them to make the CNT-composite paper. We dropped 10 μL of CNT dispersion on a silicon substrate and dried it at room temperature. We did this for each of the three different dispersions.

### 4.6. Examination of Addition of Carbon Powder

We tested the effectiveness of adding carbon powder to the CNT-composite paper used for the electrode layers of the actuator. We used CNT-composite paper with 30 mg of CNT and 30 mg of carbon powder for a circle with a diameter of 6 cm in the sample to which carbon powder was added. For comparison, we used a sample with CNT-composite paper with 30 mg of CNT and a sample of the same size with CNT-composite paper with 60 mg of CNT. For samples that did not use carbon powder, we used the same amount of catechin as the dispersant as is described in Section 4.5 and prepared them by the same procedure as is described in Section 4.5. For the sample using carbon powder, the same amount of catechin was used as the dispersant. When preparing the CNT dispersion, we added carbon powder to pure water together with CNT and a dispersant and ultrasonically dispersed it to prepare a CNT dispersion containing carbon powder. For the subsequent procedure, we performed the same procedure as that described in Section 4.5, made the samples, and measured and compared the pressure generated by each sample. In this manner, we investigated whether the addition of carbon powder to the electrode layers is effective in CNT-composite paper actuators.

## Figures and Tables

**Figure 1 molecules-26-01463-f001:**
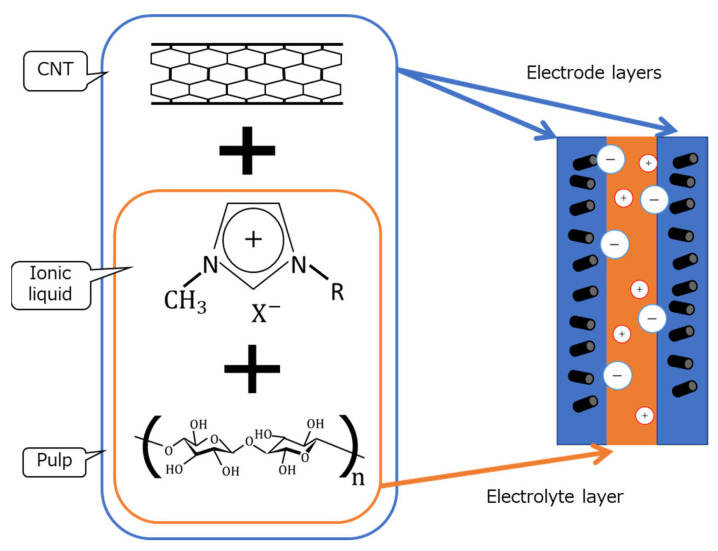
Structure of CNT-composite paper actuator.

**Figure 2 molecules-26-01463-f002:**
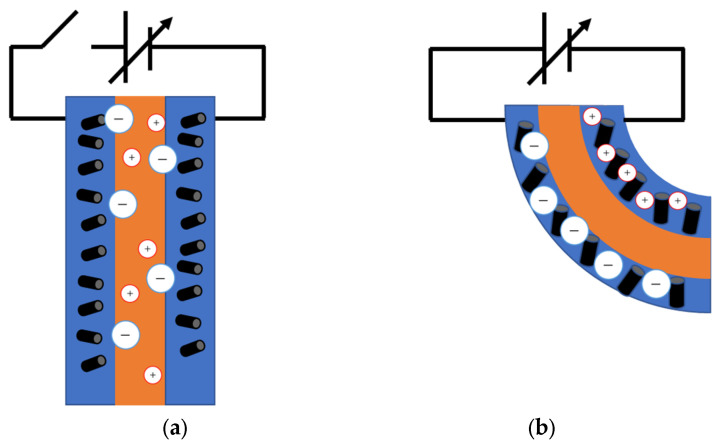
Operation of actuator. (**a**) Before applying voltage. (**b**) After applying voltage.

**Figure 3 molecules-26-01463-f003:**
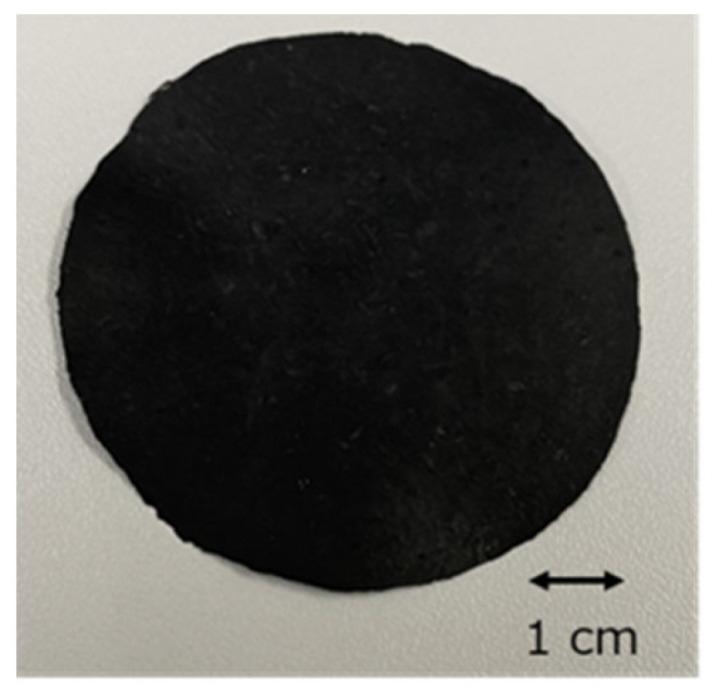
CNT-composite paper.

**Figure 4 molecules-26-01463-f004:**
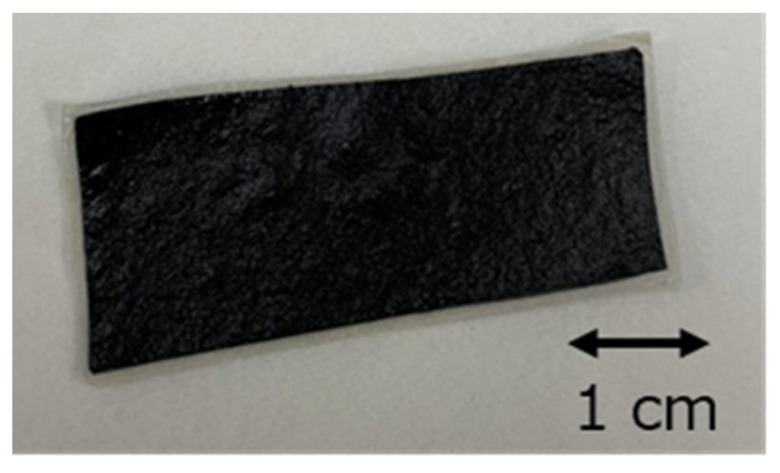
Paper actuator.

**Figure 5 molecules-26-01463-f005:**
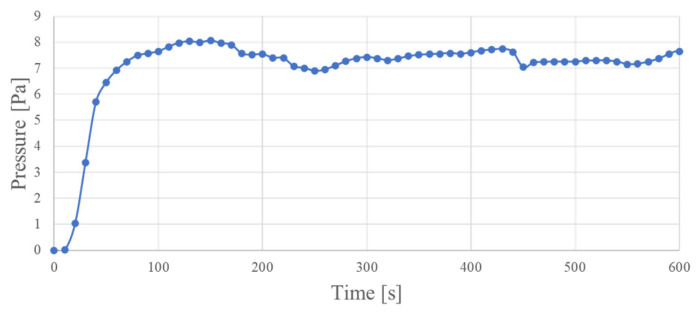
Generated pressure over time.

**Figure 6 molecules-26-01463-f006:**
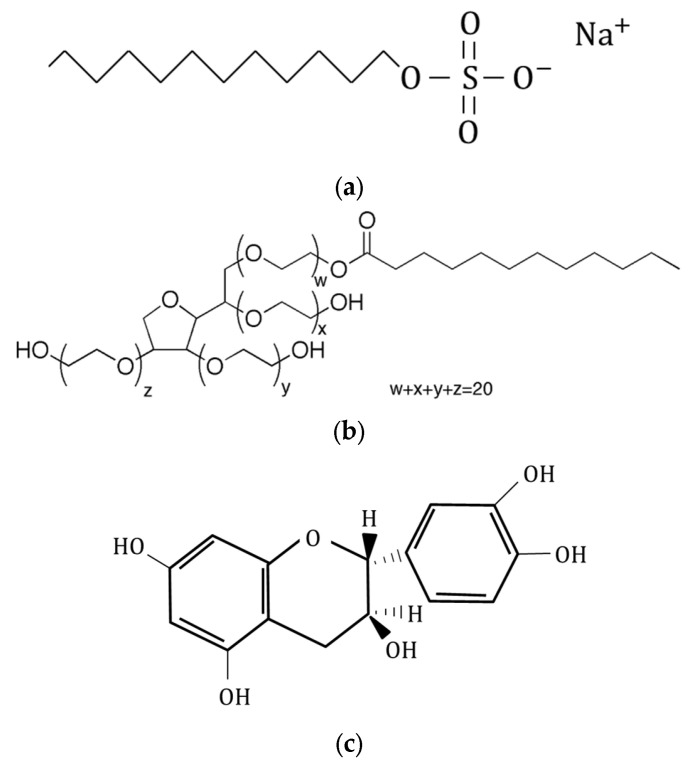
Structure of dispersants (**a**) SDS, (**b**) polysorbate 20, and (**c**) catechin.

**Figure 7 molecules-26-01463-f007:**
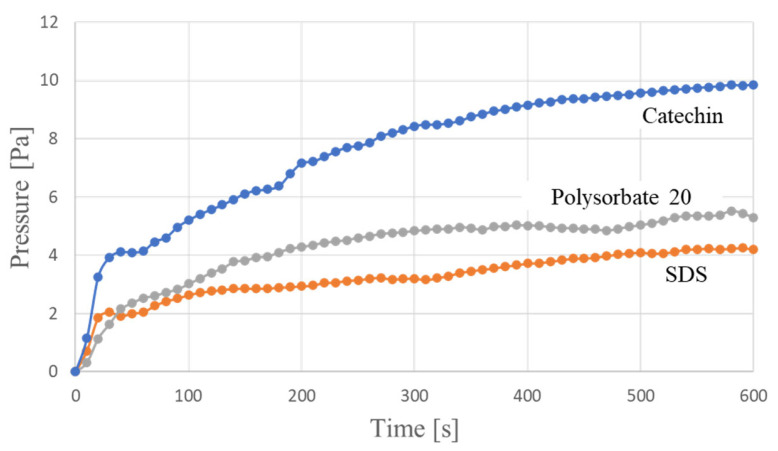
Variation over time of pressure generated with various dispersants.

**Figure 8 molecules-26-01463-f008:**
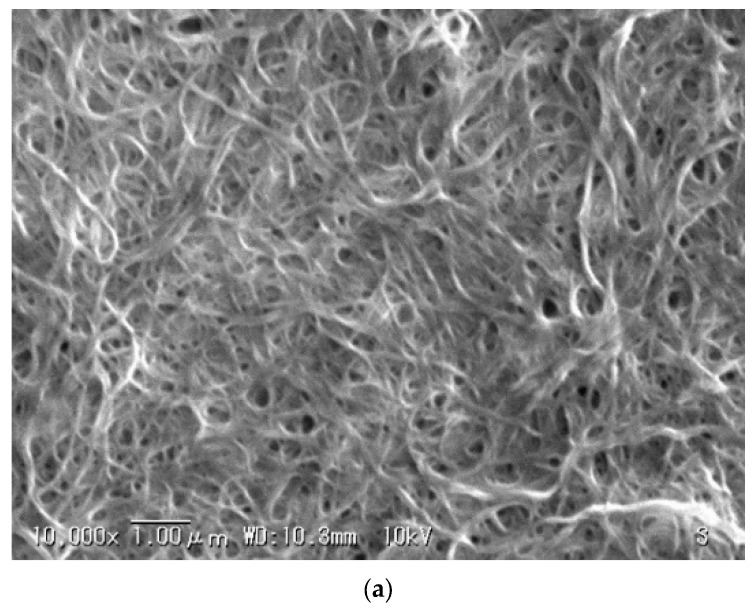
Comparison of degree of dispersion by dispersants (**a**) SDS, (**b**) polysorbate 20, and (**c**) catechin.

**Figure 9 molecules-26-01463-f009:**
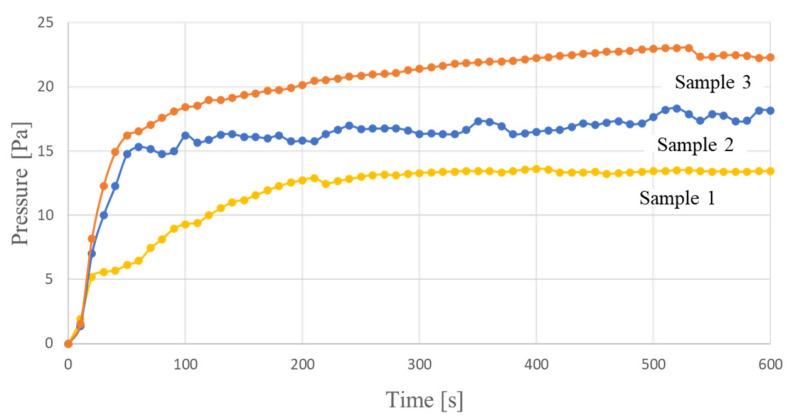
Difference in pressure with and without addition of carbon powder.

**Figure 10 molecules-26-01463-f010:**
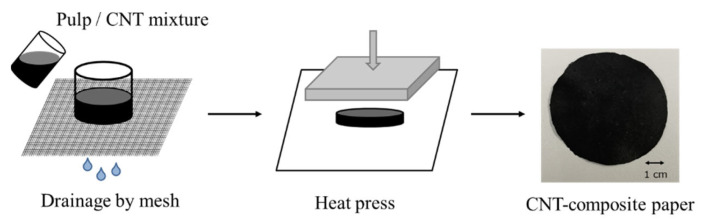
Process for making CNT-composite paper.

**Figure 11 molecules-26-01463-f011:**
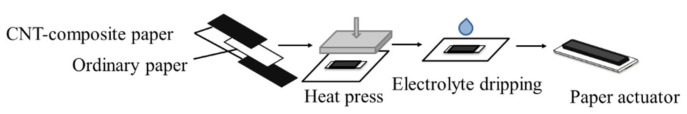
Process for making actuator.

**Table 1 molecules-26-01463-t001:** Resistance value of CNT-composite paper when each dispersant is used.

Dispersant	SDS	Polysorbate 20	Catechin
Resistance [Ω/sq]	10.7	13.8	11.9

**Table 2 molecules-26-01463-t002:** Comparison of resistance values between CNT-composite paper with added carbon powder (Sample 3) and other CNT-composite paper (Samples 1 and 2).

Sample	Sample 1	Sample 2	Sample 3
Resistance [Ω/sq]	11.3	6.22	6.98

**Table 3 molecules-26-01463-t003:** Characteristics of ionic liquid, EMI-TFSI.

	Cation	Anion
Structural formula	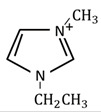	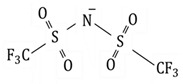
Volume [nm^3^]	0.156	0.232

## Data Availability

The data described in the manuscript are available from all authors on reasonable request.

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
