# Peer review of "Development of Paper Actuators Based on Carbon-Nanotube-Composite Paper"

_molecules, 2021, doi:10.3390/molecules26051463_

Round 1
Reviewer 1 Report
The paper presents a novel method in fabrication of paper actuators, however, the findings are not well presented make it difficult to evaluate
- It must be mentioned what type of CNT was used in the process? Diameter, length, Single or double wall, etc.
- A proofreading of paper is required, there are plethora of colloquial terms.
- An experimental approach addressing the details of manufacturing is missing.
- There is no coherent narrative in the paper and subsequent sections do not unfold logically.
- The unit of force is no Pa in Figure 5.
- It is not clear what force is being measured and how
- Scanning electron microscopy images of the microstructure is required to evaluate the dispersion of CNT.
Author Response
Response to Reviewer 1 Comments
26/Feb./2021
Dear Reviewer,
Thank you very much for your time to review and useful suggestion.
We checked all of your comments and revised our manuscript as below.
In our revised manuscript, we have highlighted additional sentences that includes answers for your comments.
We believe our revised manuscript can provide new interest for Molecules readers.
Point 1: It must be mentioned what type of CNT was used in the process? Diameter, length, Single or double wall, etc.
Response 1: Thank you very much for your pointing out. The CNTs we used in this study are ZEONANO SG101 with an average diameter of 3 to 5 nm and a length of 100 to 600 mm. We have added more information about CNTs in strings 355-356.
Point 2: A proofreading of paper is required, there are plethora of colloquial terms.
Response 2: Thank you for your comment. We understand that our English must be improved. Before submitting our paper that you reviewed, we requested native speakers of English to proofread. For this revised manuscript, we requested the proofreading again.
Point 3: An experimental approach addressing the details of manufacturing is missing.
Response 3: Thank you for your valuable feedback. We added the missing parts in the fabrication method, experimental result and discussion based on your feedback. Specifically, we added details of the pressure measurement method (subsection 4.4), confirmation of the dispersion state of the dispersions using a scanning electron microscope (subsection 2.2.1 and subsection 4.4), and consideration of the relationship between the resistance value and pressure (section 3).
Point 4: There is no coherent narrative in the paper and subsequent sections do not unfold logically.
Response 4: We modified the wording to clarify the relationship between subsection 2.2.1 and subsection 2.2.2. We changed the flow to confirm the effectiveness of the addition of carbon powder based on the results of measurement with a scanning electron microscope (strings 197-206).
Point 5: The unit of force is no Pa in Figure 5.
Response 5: Thank you very much for your pointing out. You are right. We have modified the expression from "force" to "pressure". We have made similar corrections to the following figures and related items.
Point 6: It is not clear what force is being measured and how.
Response 6: Thank you very much for your comment. As a first test, we expected the actuator to change the value of the electronic scale by pushing a piece of plastic placed on the electronic scale. We thought that this value could be regarded as the force generated by the actuator. Therefore, we derived the pressure from the value measured by the electronic scale. We have added a description of the measurement method in strings 393-400.
Point 7: Scanning electron microscopy images of the microstructure is required to evaluate the dispersion of CNT.
Response 7: Thank you for your comment. We observed the dispersions with a scanning electron microscope in a state where they were dropped on silicon substrates and dried for simplicity because it was difficult to observe the condition of CNT dispersion in the composite paper. We added the measurement results in strings 198-220. We added the specific measurement method in strings 428-439.

Reviewer 2 Report
In this study authors suggest new actuators device using CNT-composite paper. Paper is enough interesting and finally may be published in Journal.
however, some aspects not easy to evaluate: for example: authors used different concentrations of dispersants for each substance (strings 323-324) and mentioned that consentration of dispersants (strings 237-238) may be changed. Were the final concentrations selected based on the minimum resistance for each dispersant or the maximum drive force?
Even for the first original sample (whose strength characteristics are shown in Figure 5) the resistance is not clear. So aftervard it is not clear from the text of the article how the introduction of the dispersant affected the resistance as compare to the original sample.
It is really difficult to understant it there are any interrelations between resistance and drive force. When studying the effect of carbon powder additives, the samples contained catechin. (strings 344, 347). Sample with catechin (Fig. 7) and Sample 1 (Fig. 8) are different. Please explain why that samples have similar resistance, but vary in force.
Finally I think that authors should add more explanation for the experimantal results.
And one more remark - why force measured in Pa? (in all figures)? However in fig. captions it is always "Pressure". Is that Force [N] or pressure [Pa]?
Author Response
Response to Reviewer 2 Comments
26/Feb./2021
Dear Reviewer,
Thank you very much for your time to review and useful suggestion.
We checked all of your comments and revised our manuscript as below.
In our revised manuscript, we have highlighted additional sentences that includes answers for your comments.
We believe our revised manuscript can provide new interest for Molecules readers.
Point 1: Authors used different concentrations of dispersants for each substance (strings 323-324) and mentioned that concentration of dispersants (strings 237-238) may be changed. Were the final concentrations selected based on the minimum resistance for each dispersant or the maximum drive force?
Response 1: Thank you very much for your pointing out. That was misleading explanations. We believe that the concentration of the dispersant can be related to the obtained resistance value. To clarity this, we tested each concentration described in the explanation for three dispersants. We set the amount of each dispersant so that even if the amount was increased further, almost no decrease in resistance was confirmed. We have not yet verified whether the pressure generated is the maximum, so we think that further investigation is needed in the future. We added the explanation about this in strings 416-418.
Point 2: Even for the first original sample (whose strength characteristics are shown in Figure 5) the resistance is not clear. So afterward it is not clear from the text of the article how the introduction of the dispersant affected the resistance as compare to the original sample.
Response 2: Thank you very much. As you mentioned, that was unclear. This time, we have shown a sample using catechin as a dispersant as a representative. We added this to subsection 2.1. We made this sample under the same conditions as the sample with catechin (Fig. 7) and sample 1 (Fig. 8). The difference in the results is considered to be due to the variation caused by the manufacturing process. We have added the dispersant and resistance values of the measured sample in strings 116-117.
Point 3: It is really difficult to understand it there are any interrelations between resistance and drive force. When studying the effect of carbon powder additives, the samples contained catechin. (strings 344, 347). Sample with catechin (Fig. 7) and Sample 1 (Fig. 8) are different. Please explain why that samples have similar resistance, but vary in force.
Response 3: Thank you very much for your comments. It is very important point. However, the explanation was not enough. In this sample, the paper was handmade by following the paper making method, so we think that the results differed due to factors such as variations in the structure of the paper even in samples with similar resistance values. We consider such variations in the manufacturing process as a topic for future study. However, the differences of magnitude of pressures may depend on the contained bundled CNTs as apacers for ions, we considered. We added this in strings 327-336.
Point 4: Finally I think that authors should add more explanation for the experimental results.
Response 4: Thank you for your valuable feedback. We have added results and discussions as described above based on your feedback.
Point 5: And one more remark - why force measured in Pa? (in all figures)? However in fig. captions it is always "Pressure". Is that Force [N] or pressure [Pa]?
Response 5: Thank you very much for your pointing out. That was misleading expressions. This time, we evaluated the performance of the actuator by pressure. Therefore, we have revised the expression from "force" to "pressure" in the figures and related items.

Round 2
Reviewer 1 Report
The applied changes are appropriate and the paper is recommended for publication in its current form.